# ROBUSTIFY TRANSFORMERS WITH ROBUST KERNEL DENSITY ESTIMATION

## ABSTRACT

Recent advances in Transformer architecture have empowered its empirical success in various tasks across different domains. However, existing works mainly focus on improving the standard accuracy and computational cost, without considering the robustness of contaminated samples. Existing work (Nguyen et al., 2022) has shown that the self-attention mechanism, which is the center of the Transformer architecture, can be viewed as a non-parametric estimator based on the well-known kernel density estimation (KDE). This motivates us to leverage a set of robust kernel density estimation methods in the self-attention mechanism, to alleviate the issue of the contamination of data by down-weighting the weight of bad samples in the estimation process. The modified self-attention mechanism can be incorporated into different Transformer variants. Empirical results on language modeling and image classification tasks demonstrate the effectiveness of this approach.

## 1 INTRODUCTION

Attention mechanisms and transformers (Vaswani et al., 2017) have been widely used in machine learning community (Lin et al., 2021; Tay et al., 2020; Khan et al., 2021). Transformer-based models are now among the best deep learning architectures on a variety of applications, including those in natural language processing (Devlin et al., 2019; Al-Rfou et al., 2019; Dai et al., 2019; Child et al., 2019; Raffel et al., 2020; Baevski & Auli, 2019; Brown et al., 2020; Dehghani et al., 2019), computer vision (Dosovitskiy et al., 2021; Liu et al., 2021; Touvron et al., 2021a; Ramesh et al., 2021; Radford et al., 2021; Fan et al., 2021; Liu et al., 2022), and reinforcement learning (Chen et al., 2021; Janner et al., 2021). Transformers have also been well-known for their effectiveness in transferring knowledge from pretraining tasks to downstream applications with weak supervision or no supervision (Radford et al., 2018; 2019; Devlin et al., 2019; Yang et al., 2019; Liu et al., 2019).

**Contribution** Despite having an appealing performance, the robustness of the conventional attention module still remains an open question in the literature. In this paper, to robustify the attention mechanism and transformer models, we first revisit the interpretation of the self-attention in the transformer as the Nadaraya-Watson (NW) estimator (Nadaraya, 1964) in a non-parametric regression problem in the recent work of Nguyen et al. (2022). Putting in the context of transformer, the NW estimator is constructed mainly based on the kernel density estimators (KDE) of the keys and queries. However, the KDE is not robust to the outliers (Kim & Scott, 2012), which leads to the robustness issue of the NW estimator and the self-attention in transformer when there are outliers in the data. To improve the robustness of the KDE, we first show that the KDE can be viewed as an optimal solution of the kernel regression problem in the reproducing kernel Hilbert space (RKHS). Then, to robustify the KDE, we can either robustify the loss function of the kernel regression problem via some robust loss functions, such as the well-known Huber loss function (Huber, 1992), or reweight the contaminated densities via scaling and projecting the original densities. The family of robust KDE can be used to construct a set of novel robust attentions in transformer, which also improves the robustness issue of the transformer. In summary, our contribution is two-fold:

- By connecting the dot-product self-attention mechanism in transformer with the nonparametric kernel regression problem in reproducing kernel Hilbert space (RKHS), we propose a novel robust transformer framework, based on replacing the dot-product attention by an attention arising from a set of robust kernel density estimators associated with the robust

kernel regression problem. Comparing to the standard soft-max transformer, the family of robustified transformers only requires computing an extra set of weights.

- Extensive experiments on both vision and language modeling tasks demonstrate that our proposed framework has favorable performance under various attacks. Furthermore, the proposed robust transformer framework is flexible and can be incorporated into different Transformer variants.

**Organization**    The paper is organized as follows. In Section 2, we provide background on self-attention mechanism in Transformer and its connection to the Nadaraya-Watson (NW) estimator in the nonparametric regression problem, which can be constructed via KDE. In Section 3, we first connect the KDE to a kernel regression problem in the reproducing kernel Hilbert space (RKHS) and demonstrate that it is not robust to the outliers. Then, we construct the robust self-attention mechanism for the Transformer by leveraging a set of robust KDE methods. We empirically validate the advantage of the proposed robust self-attension mechanism, over the standard softmax transformer along with other baselines over both language modeling and image classification tasks in Section 4. Finally, we discuss the related works in Section 5 while conclude the paper in Section 6.

## 2   BACKGROUND: SELF-ATTENTION MECHANISM FROM A NON-PARAMETRIC REGRESSION PERSPECTIVE

In this section, we first provide background on the self-attention mechanism in transformer in Section 2. We then revisit the connection between the self-attention and the Nadaraya-Watson estimator in a nonparametric regression problem in Section 2.2.

### 2.1   SELF-ATTENTION MECHANISM

Given an input sequence $\boldsymbol{X} = [\boldsymbol{x}_1, \dots, \boldsymbol{x}_N]^\top \in \mathbb{R}^{N \times D_x}$ of $N$ feature vectors, the self-attention transforms it into another sequence $\mathbf{H} := [\boldsymbol{h}_1, \cdots, \boldsymbol{h}_N]^\top \in \mathbb{R}^{N \times D_v}$ as follows:

$$\boldsymbol{h}_i = \sum_{j \in [N]} \text{softmax}\Big(\frac{\boldsymbol{q}_i^\top \boldsymbol{k}_j}{\sqrt{D}}\Big)\boldsymbol{v}_j, \text{ for } i = 1, \dots, N, \tag{1}$$

where the scalar $\text{softmax}((\boldsymbol{q}_i^\top \boldsymbol{k}_j)/\sqrt{D})$ can be understood as the attention $\boldsymbol{h}_i$ pays to the input feature $\boldsymbol{x}_j$. The vectors $\boldsymbol{q}_i, \boldsymbol{k}_j$, and $\boldsymbol{v}_j$ are the query, key, and value vectors, respectively, and are computed as follows:

$$
\begin{aligned}
{[\boldsymbol{q}_1, \boldsymbol{q}_2, \dots, \boldsymbol{q}_N]}^\top &:= \boldsymbol{Q} = \boldsymbol{X}\boldsymbol{W}_Q^\top \in \mathbb{R}^{N \times D}, \\
{[\boldsymbol{k}_1, \boldsymbol{k}_2, \dots, \boldsymbol{k}_N]}^\top &:= \boldsymbol{K} = \boldsymbol{X}\boldsymbol{W}_K^\top \in \mathbb{R}^{N \times D}, \\
{[\boldsymbol{v}_1, \boldsymbol{v}_2, \dots, \boldsymbol{v}_N]}^\top &:= \boldsymbol{V} = \boldsymbol{X}\boldsymbol{W}_V^\top \in \mathbb{R}^{N \times D_v},
\end{aligned}
\tag{2}
$$

where $\boldsymbol{W}_Q, \boldsymbol{W}_K \in \mathbb{R}^{D \times D_x}, \boldsymbol{W}_V \in \mathbb{R}^{D_v \times D_x}$ are the weight matrices. Equation 1 can be written as:

$$\mathbf{H} = \text{softmax}\Big(\frac{\boldsymbol{Q}\boldsymbol{K}^\top}{\sqrt{D}}\Big)\boldsymbol{V}, \tag{3}$$

where the softmax function is applied to each row of the matrix $(\boldsymbol{Q}\boldsymbol{K}^\top)/\sqrt{D}$. equation 3 is also called the "softmax attention". For each query vector $\boldsymbol{q}_i$ for $i = 1, \cdots, N$, an equivalent form of equation 3 to compute the output vector $\boldsymbol{h}_i$ is given by

$$\boldsymbol{h}_i = \sum_{j \in [N]} \text{softmax}\Big(\frac{\boldsymbol{q}_i^\top \boldsymbol{k}_j}{\sqrt{D}}\Big)\mathbf{v}_j := \sum_{j \in [N]} a_{ij}\mathbf{v}_j. \tag{4}$$

In this paper, we call a transformer built with softmax attention standard transformer or transformer.

### 2.2   A NON-PARAMETRIC REGRESSION PERSPECTIVE OF SELF-ATTENTION

We now review the connection between the self-attention mechanism in equation 4 and the nonparametric regression, which has been discussed in the recent work (Nguyen et al., 2022). Assume

we have the key and value vectors $\{\boldsymbol{k}_j, \mathbf{v}_j\}_{j\in[N]}$ that is collected from the following data generating process:

$$\mathbf{v} = f(\boldsymbol{k}) + \varepsilon, \tag{5}$$

where $\varepsilon$ is some noise vectors with $\mathbb{E}[\varepsilon] = 0$, and $f$ is the unknown function that we want to estimate. We consider a random design setting where the key vectors $\{\boldsymbol{k}_j\}_{j\in[N]}$ are i.i.d. samples from the distribution $p(\boldsymbol{k})$, and we use $p(\mathbf{v}, \boldsymbol{k})$ to denote the joint distribution of $(\mathbf{v}, \boldsymbol{k})$ defined by equation 5. Our target is to estimate $f(\boldsymbol{q})$ for any new queries $\boldsymbol{q}$.

Nadaraya (1964) provides a non-parametric approach to estimate the function $f$, which is known as the the Nadaraya-Watson (NW) estimator, the kernel regression estimator or the local constant estimator. The main idea of the NW estimator is that

$$f(\boldsymbol{k}) = \mathbb{E}[\mathbf{v}|\boldsymbol{k}] = \int_{\mathbb{R}^D} \mathbf{v} \cdot p(\mathbf{v}|\boldsymbol{k})d\mathbf{v} = \int_{\mathbb{R}^D} \frac{\mathbf{v} \cdot p(\mathbf{v}, \boldsymbol{k})}{p(\boldsymbol{k})}d\mathbf{v}, \tag{6}$$

where the first equation comes from the fact that $\mathbb{E}[\varepsilon] = 0$, the second equation comes from the definition of conditional expectation and the last inequality comes from the definition of the conditional density. With equation 6, we know, to provide an estimation of $f$, we just need to obtain estimations for both the joint density function $p(\mathbf{v}, \boldsymbol{k})$ and the marginal density function $p(\boldsymbol{k})$. One of the most popular approaches for the density estimation problem is the kernel density estimation (KDE) (Rosenblatt, 1956; Parzen, 1962), which requires a kernel $k_\sigma$ with the bandwidth parameter $\sigma$ satisfies $\int_{\mathbb{R}^D} k_\sigma(\boldsymbol{x} - \boldsymbol{x}')d\boldsymbol{x} = 1, \forall \boldsymbol{x}'$, and estimate the density as

$$\hat{p}_\sigma(\mathbf{v}, \boldsymbol{k}) = \frac{1}{N} \sum_{j\in[N]} k_\sigma\left([\mathbf{v}, \boldsymbol{k}] - [\mathbf{v}_j, \boldsymbol{k}_j]\right), \quad \hat{p}_\sigma(\boldsymbol{k}) = \frac{1}{N} \sum_{j\in[N]} k_\sigma(\boldsymbol{k} - \boldsymbol{k}_j), \tag{7}$$

where $[\mathbf{v}, \boldsymbol{k}]$ denotes the concatenation of $\mathbf{v}$ and $\boldsymbol{k}$. Specifically, when $k_\sigma$ is the isotropic Gaussian kernel $k_\sigma(\boldsymbol{x} - \boldsymbol{x}') = \exp\left(-\|\boldsymbol{x} - \boldsymbol{x}'\|^2/(2\sigma^2)\right)$, we have

$$\hat{p}_\sigma(\mathbf{v}, \boldsymbol{k}) = \frac{1}{N} \sum_{j\in[N]} k_\sigma(\mathbf{v} - \mathbf{v}_j)k_\sigma(\boldsymbol{k} - \boldsymbol{k}_j). \tag{8}$$

Given the kernel density estimators equation 7 and equation 8, as well as the formulation in equation 6, we obtain the NW estimator of the function $f$:

$$
\begin{aligned}
\widehat{f}_\sigma(\boldsymbol{k}) &= \int_{\mathbb{R}^D} \frac{\mathbf{v} \cdot \hat{p}_\sigma(\mathbf{v}, \boldsymbol{k})}{\hat{p}_\sigma(\boldsymbol{k})}d\mathbf{v} = \int_{\mathbb{R}^D} \frac{\mathbf{v} \cdot \sum_{j\in[N]} k_\sigma(\mathbf{v} - \mathbf{v}_j)k_\sigma(\boldsymbol{k} - \boldsymbol{k}_j)}{\sum_{j\in[N]} k_\sigma(\boldsymbol{k} - \boldsymbol{k}_j)}d\mathbf{v} \\
&= \frac{\sum_{j\in[N]} k_\sigma(\boldsymbol{k} - \boldsymbol{k}_j) \int \mathbf{v} \cdot k_\sigma(\mathbf{v} - \mathbf{v}_j)d\mathbf{v}}{\sum_{j\in[N]} k_\sigma(\boldsymbol{k} - \boldsymbol{k}_j)} \\
&= \frac{\sum_{j\in[N]} \mathbf{v}_j k_\sigma(\boldsymbol{k} - \boldsymbol{k}_j)}{\sum_{j\in[N]} k_\sigma(\boldsymbol{k} - \boldsymbol{k}_j)}.
\end{aligned} \tag{9}
$$

Now we show how the self-attention mechanism is related to the NW estimator. Note that

$$
\begin{aligned}
\widehat{f}_\sigma(\boldsymbol{q}) &= \frac{\sum_{j\in[N]} \mathbf{v}_j \exp\left(-\|\boldsymbol{q} - \boldsymbol{k}_j\|^2/2\sigma^2\right)}{\sum_{j\in[N]} \exp\left(-\|\boldsymbol{q} - \boldsymbol{k}_j\|^2/2\sigma^2\right)} \\
&= \frac{\sum_{j\in[N]} \mathbf{v}_j \exp\left[-\left(\|\boldsymbol{q}\|^2 + \|\boldsymbol{k}_j\|^2\right)/2\sigma^2\right] \exp\left(\boldsymbol{q}^\top \boldsymbol{k}_j/\sigma^2\right)}{\sum_{j\in[N]} \exp\left[-\left(\|\boldsymbol{q}\|^2 + \|\boldsymbol{k}_j\|^2\right)/2\sigma^2\right] \exp\left(\boldsymbol{q}^\top \boldsymbol{k}_j/\sigma^2\right)}.
\end{aligned} \tag{10}
$$

If the keys $\{\boldsymbol{k}_j\}_{j\in[N]}$ are normalized, we can further simplify $\widehat{f}_\sigma(\boldsymbol{q}_i)$ in equation 9 to

$$\widehat{f}_\sigma(\boldsymbol{q}_i) = \frac{\sum_{j\in[N]} \mathbf{v}_j \exp\left(\boldsymbol{q}\boldsymbol{k}_j^\top/\sigma^2\right)}{\sum_{j\in[N]} \exp\left(\boldsymbol{q}\boldsymbol{k}_j^\top/\sigma^2\right)} = \sum_{j\in[N]} \mathrm{softmax}\left(\boldsymbol{q}^\top \boldsymbol{k}_j/\sigma^2\right) \mathbf{v}_j. \tag{11}$$

Such an assumption on the normalized key $\{\boldsymbol{k}_j\}_{j\in[N]}$ can be mild, as in practice we always have an normalization step on the key to stabilize the training of the transformer (Schlag et al., 2021). If we choose $\sigma^2 = \sqrt{D}$, where $D$ is the dimension of $\boldsymbol{q}$ and $\boldsymbol{k}_j$, then $\widehat{f}_\sigma(\boldsymbol{q}_i) = \boldsymbol{h}_i$. As a result, the self-attention mechanism in fact performs a non-parametric regression with NW-estimator and isotropic Gaussian kernel when the keys are normalized.

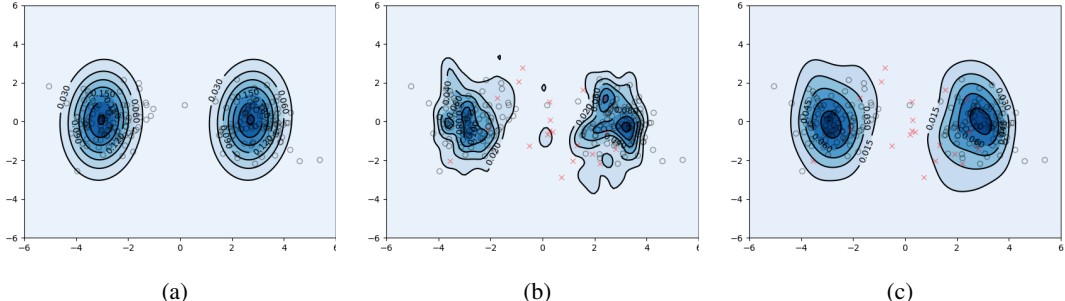

Figure 1: Contour plots of density estimation of the 2-dimensional query vector embedding in an attention layer of the transformer when using (b) KDE (equation 12) and (c) RKDE (equation 13) with Huber loss (equation 14), where (a) is the true density function. We draw 1000 samples (gray circles) from a multivariate normal density and 100 outliers (red cross) from a gamma distribution as the contaminating density. RKDE can be less affected by outliers when computing self-attention as nonparametric regression.

## 3   ROBUSTIFY TRANSFORMER WITH ROBUST KERNEL DENSITY ESTIMATION

As we have seen in Section 2, the self-attention mechanism can be interpreted as an NW estimator for the unknown function where the density is estimated with KDE using the isotropic Gaussian kernel. In this section, we first re-interpret KDE as a regression in the Reproducing Kernel Hilbert Space (RKHS), which shows that the vanilla KDE is sensitive to the data corruption. Instead, we observe that, variants of the kernel density estimation such as robust KDE (Kim & Scott, 2012) and scaled projection KDE (Vandermeulen & Scott, 2014), can down-weight the importance of the potential corrupted data and obtain a robust density estimator. Based on the variants, we derive the corresponding robust version of the NW-estimator, and show how to use this to replace the self-attention mechanism, and eventually lead to a more robust Transformer variants.

### 3.1   KDE AS A REGRESSION PROBLEM IN RKHS

We start from the formal definition of the RKHS. The space $\mathcal{H}_k = \{f \mid f : \mathcal{X} \to \mathbb{R}\}$ is called an RKHS associated with the kernel $k$, where $k : \mathcal{X} \times \mathcal{X} \to \mathbb{R}$, if it is a Hilbert space with the following two properties: (1) $k(\boldsymbol{x}, \cdot) \in \mathcal{H}_k, \forall \boldsymbol{x} \in \mathcal{X}$; (2) the reproducing property: $\forall f \in \mathcal{H}$, $f(\boldsymbol{x}) = \langle f, k(\boldsymbol{x}, \cdot) \rangle_{\mathcal{H}_k}$, where $\langle \cdot, \cdot \rangle_{\mathcal{H}_k}$ denotes the RKHS inner product. With slightly abuse of notation, we define $k_\sigma(\boldsymbol{x}, \boldsymbol{x}') = k_\sigma(\boldsymbol{x} - \boldsymbol{x}')$. By the definition of the RKHS and the KDE estimator, we know $\hat{p}_\sigma = \frac{1}{N} \sum_{j \in [N]} k_\sigma(\boldsymbol{x}_j, \cdot) \in \mathcal{H}_{k_\sigma}$. In fact, $\hat{p}_\sigma$ is the optimal solution of the following least-square regression problem in RKHS:

$$\hat{p}_\sigma = \underset{p \in \mathcal{H}_{k_\sigma}}{\arg\min} \sum_{j \in [N]} \frac{1}{N} \|k_\sigma(\boldsymbol{x}_j, \cdot) - p\|^2_{\mathcal{H}_{k_\sigma}} . \tag{12}$$

Note that, in equation 12, we have the same weight $1/N$ on each of the error $\|k_\sigma(\boldsymbol{x}_j, \cdot) - p\|^2_{\mathcal{H}_{k_\sigma}}$. This works well if there are no outliers in $\{k_\sigma(\boldsymbol{x}_j, \cdot)\}_{j \in [N]}$. However, when we have outliers (e.g., when there exists some $j$, such that $\|k_\sigma(\boldsymbol{x}_j, \cdot)\|_{\mathcal{H}_{k_\sigma}} \gg \|k_\sigma(\boldsymbol{x}_i, \cdot)\|_{\mathcal{H}_{k_\sigma}}, \forall i \in [N], i \neq j$), the error on the outliers will dominate the whole error and lead to substantially worse estimation on the entire density. We illustrate the robustness issue of the KDE in Figure 1.

Combining the viewpoint that KDE is not robust to outliers with the interpretation of section 2.2 implies that the transformer is also not robust when there are outliers in the data. The robustness issue of transformer has mostly been studied in the vision domain, such as (Mahmood et al., 2021; Mao et al., 2022; Zhou et al., 2022). These works modify the original architectures of vision transformer and introduces extra parameters. A representative one is Mao et al. (2022), which proposed position-based attention by adding on another fully connected layer. However, this approach will cause bi-directional information flow for positional-sensitive dataset such as text or sequences and is therefore limited to image data. We take a different view of the robustness problem in the RKHS domain and provide a unified framework for different data modalities.

## 3.2 ROBUST KDE

Motivated by the robust regression (Fox & Weisberg, 2002), Kim & Scott (2012) proposed a robust version of KDE, by replacing the least-square loss in equation 12 with a robust loss function $\rho$:

$$\hat{p}_{\text{robust}} = \arg\min_{p \in \mathcal{H}_{k_\sigma}} \sum_{j \in [N]} \rho\left(\|k_\sigma(\boldsymbol{x}_j, \cdot) - p\|_{\mathcal{H}_{k_\sigma}}\right). \tag{13}$$

Examples of the robust loss functions $\rho$ include the Huber loss (Huber, 1992), Hampel loss (Hampel et al., 1986), Welsch loss (Welsch & Becker, 1975) and Tukey loss (Fox & Weisberg, 2002). We empirically evaluate different loss functions in our experiments. For simplicity, we use the Huber loss function as the demonstrating example, which is defined as follows:

$$\rho(x) := \left\{ \begin{array}{ll} x^2/2, & 0 \leq x \leq a \\ ax - a^2/2, & a < x, \end{array} \right. \tag{14}$$

where $a$ is a constant. Kim & Scott (2012) shows the solution of this robust regression problem has the following form:

**Proposition 1.** *Assume the robust loss function $\rho$ is non-decreasing in $[0, \infty]$, $\rho(0) = 0$ and $\lim_{x \to 0} \frac{\rho(x)}{x} = 0$. Define $\psi(x) := \frac{\rho'(x)}{x}$ and assume $\psi(0) = \lim_{x \to 0} \frac{\rho'(x)}{x}$ exists and finite. Then the optimal $\hat{p}_{robust}$ can be written as*

$$\hat{p}_{robust} = \sum_{j \in [N]} \omega_j k_\sigma(\boldsymbol{x}_j, \cdot),$$

*where $\omega = (\omega_1, \cdots, \omega_N) \in \Delta_N$, and $\omega_j \propto \psi\left(\|k_\sigma(\boldsymbol{x}_j, \cdot) - \hat{p}_{robust}\|_{\mathcal{H}_{k_\sigma}}\right)$. Here $\Delta_n$ denotes the $n$-dimensional simplex.*

The proof of this proposition can be found in Appendix A. For Huber loss function, we have that

$$\psi(x) := \left\{ \begin{array}{ll} 1, & 0 \leq x \leq a \\ a/x, & a < x. \end{array} \right.$$

Hence, when the error $\|k_\sigma(\boldsymbol{x}_j, \cdot), \cdot - \hat{p}_{\text{robust}}\|_{\mathcal{H}_{k_\sigma}}$ is over the threshold $a$, the final estimator will down-weight the importance of $k_\sigma(\boldsymbol{x}_j, \cdot)$. This is in sharp contrast with the standard KDE method, which will assign uniform weights to all of the $k_\sigma(\boldsymbol{x}_j, \cdot)$. One additional issue is that, the estimator provided in Proposition 1 is circularly defined, as $\hat{p}_{\text{robust}}$ is defined via $\omega$, and $\omega$ depends on $\hat{p}_{\text{robust}}$. To address this issue, Kim & Scott (2012) proposed to estimate $\omega$ with an iterative algorithm termed as kernelized iteratively re-weighted least-squares (KIRWLS) algorithm. The algorithm starts with some randomly initialized $\omega^{(0)} \in \Delta_n$, and perform the following iterative updates:

$$\hat{p}_{\text{robust}}^{(k)} = \sum_{j \in [N]} \omega_i^{(k-1)} k_\sigma(\boldsymbol{x}_j, \cdot), \quad \omega_j^{(k)} = \frac{\psi\left(\left\|k_\sigma(\boldsymbol{x}_j, \cdot) - \hat{p}_{\text{robust}}^{(k)}\right\|_{\mathcal{H}_{k_\sigma}}\right)}{\sum_{j \in [N]} \psi\left(\left\|k_\sigma(\boldsymbol{x}_j, \cdot) - \hat{p}_{\text{robust}}^{(k)}\right\|_{\mathcal{H}_{k_\sigma}}\right)}. \tag{15}$$

Note that, the optimal $\hat{p}_{\text{robust}}$ is the fixed point of this iterative updates, and Kim & Scott (2012) shows that the proposed algorithm converges under standard regularity conditions. Furthermore, one can directly compute the term $\left\|k_\sigma(\boldsymbol{x}_j, \cdot) - \hat{p}_{\text{robust}}^{(k)}\right\|_{\mathcal{H}_{k_\sigma}}$ via the reproducing property:

$$\left\|k_\sigma(\boldsymbol{x}_j, \cdot) - \hat{p}_{\text{robust}}^{(k)}\right\|_{\mathcal{H}_{k_\sigma}}^2 = \langle k_\sigma(\boldsymbol{x}_j, \cdot), k_\sigma(\boldsymbol{x}_j, \cdot) \rangle_{\mathcal{H}_{k_\sigma}} - 2\left\langle k_\sigma(\boldsymbol{x}_j, \cdot), \hat{p}_{\text{robust}}^{(k)} \right\rangle_{\mathcal{H}_{k_\sigma}} + \left\langle \hat{p}_{\text{robust}}^{(k)}, \hat{p}_{\text{robust}}^{(k)} \right\rangle_{\mathcal{H}_{k_\sigma}}$$

$$= k_\sigma(\boldsymbol{x}_j, \boldsymbol{x}_j) - 2 \sum_{m \in [N]} \omega_m^{(k-1)} k_\sigma(\boldsymbol{x}_m, \boldsymbol{x}_j)$$

$$+ \sum_{m \in [N], n \in [N]} \omega_m^{(k-1)} \omega_n^{(k-1)} k_\sigma(\boldsymbol{x}_m, \boldsymbol{x}_n).$$

Therefore, the weights can be updated without mapping the data to the Hilbert space.

---

**Algorithm 1** Procedure of Computing Attention Vector of Transformer-RKDE/SPKDE

---

1: **Input**: $\mathbf{Q} = \{\boldsymbol{q}_i\}_{i \in [N]}$, $\mathbf{K} = \{\boldsymbol{k}_j\}_{j \in [N]}$, $\mathbf{V} = \{\mathbf{v}_l\}_{l \in [N]}$, initial weights $\omega^{(0)}$
2: Normalize $\mathbf{K} = \{\boldsymbol{k}_j\}_{j \in [N]}$ along the head dimension.
3: Compute kernel function between each pair of sequence: $k_\sigma(\mathbf{Q}, \mathbf{K}) = \{k_\sigma(\boldsymbol{q}_i - \boldsymbol{k}_j)\}_{i,j \in [N]}$.
4: (Optional) apply attention mask on $k_\sigma(\mathbf{Q}, \mathbf{K})$.
5: [RKDE] Update weights $\omega^{(0)}$ for marginal/joint density by $\omega_j^{(1)} = \dfrac{\psi\left(\left\|k_\sigma(\boldsymbol{k}_j, \cdot) - \hat{p}_{\text{robust}}^{(k)}(\boldsymbol{k})\right\|_{\mathcal{H}_{k_\sigma}}\right)}{\sum_{j \in [N]} \psi\left(\left\|k_\sigma(\boldsymbol{k}_j, \cdot) - \hat{p}_{\text{robust}}^{(k)}(\boldsymbol{k})\right\|_{\mathcal{H}_{k_\sigma}}\right)}$.
6: [SPKDE] Obtain optimal weights for marginal/joint density via solving equation 17.
7: Obtain attention vector via robust self-attention $\widehat{\boldsymbol{h}}_i = \dfrac{\sum_{j \in [N]} \mathbf{v}_j \omega_j^{\text{joint}} k_\sigma(\boldsymbol{q}_i - \boldsymbol{k}_j)}{\sum_{j \in [N]} \omega_j^{\text{marginal}} k_\sigma(\boldsymbol{q}_i - \boldsymbol{k}_j)}$.

---

**Scaled Projection KDE (SPKDE)** Vandermeulen & Scott (2014) is one other option of robust KDE in the RKHS space. It essentially scale the original KDE and project it to its nearest weighted KDE in the $L_2$ norm. The resulting weighted KDE can allocate more weight to high density regions and truncate the weights for anomalous samples. Specifically, given the scaling factor $\beta > 1$, and let $\mathcal{C}_\sigma^N$ be the convex hull of $k_\sigma(x_1, \cdot), \dots, k_\sigma(x_N, \cdot) \in \mathcal{H}_{k_\sigma}$, i.e., the space of weighted KDEs, the optimal density $\hat{p}_{\text{robust}}$ is given by

$$\hat{p}_{\text{robust}} = \arg\min_{p \in \mathcal{C}_\sigma^N} \|\frac{\beta}{N} \sum_{j \in [N]} k_\sigma(x_j, \cdot) - p\|_{\mathcal{H}_{k_\sigma}}^2, \tag{16}$$

which is guaranteed to have a unique minimizer since we are projecting in a Hilbert space and $\mathcal{C}_\sigma^N$ is closed and convex. Note that, $\hat{p}_{\text{robust}}$ can also be represented as $\hat{p}_{\text{robust}} = \sum_{j \in [N]} \omega_j k_\sigma(x_j, \cdot)$, $\omega \in \Delta^N$, which is similar to robust KDE by Kim & Scott (2012). Then equation 16 can be written as a quadratic programming (QP) problem over $\omega$. Let $G$ be the Gram matrix of $k_\sigma$ and $q = G\mathbf{1}\frac{\beta}{N}$, then the QP can be written as follows

$$\min_\omega \omega^\top G \omega - 2q^\top \omega, \quad \text{subject to } \omega \in \Delta^N. \tag{17}$$

Since the Gram matrix $G$ is defined to be positive-semidefinite, this QP is convex. In practice, one can leverage commonly used solvers to efficiently obtain the solution and the optimal density $\hat{p}_{\text{robust}}$.

### 3.3 ROBUST SELF-ATTENTION MECHANISM

Now we describe the robust self-attention mechanism we use. We consider the density estimator of the joint distribution and the marginal distribution from the robust KDE:

$$\hat{p}_{\text{robust}}(\mathbf{v}, \boldsymbol{k}) = \sum_{j \in [N]} \omega_j^{\text{joint}} k_\sigma([\mathbf{v}_j, \boldsymbol{k}_j], [\mathbf{v}, \boldsymbol{k}]), \quad \hat{p}_{\text{robust}} = \sum_{j \in [N]} \omega_j^{\text{marginal}} k_\sigma(\boldsymbol{k}_j, \boldsymbol{k}).$$

With the similar computation, the robust self-attention mechanism we use is defined as

$$\widehat{\boldsymbol{h}}_i = \frac{\sum_{j \in [N]} \mathbf{v}_j \omega_j^{\text{joint}} k_\sigma(\boldsymbol{q}_i - \boldsymbol{k}_j)}{\sum_{j \in [N]} \omega_j^{\text{marginal}} k_\sigma(\boldsymbol{q}_i - \boldsymbol{k}_j)}, \tag{18}$$

where $\omega^{\text{joint}}$ and $\omega^{\text{marginal}}$ are obtained via either the KIRWLS algorithm or results from the QP solver. We term the transformer models that employ robust KDE and SPKDE as Transformer-RKDE and Transformer-SPKDE, respectively. We will show in our experiments on language modeling and image classification that SPKDE performs better empirically as it finds the optimal set of weights.

*Remark* 1. Note that, the computation of $\{\omega_j^{\text{marginal}}\}_{j \in [N]}$ and $\{\omega_j^{\text{joint}}\}_{j \in [N]}$ are separate as $\omega_j^{\text{joint}}$ involves both keys and values vectors. During the empirical evaluation, we concatenate the keys and values along the head dimension to obtain the weights for the joint density $\hat{p}_{\text{robust}}(\mathbf{v}, \boldsymbol{k})$ and only use the key vectors for obtaining the set of weights for the marginal $\hat{p}_{\text{robust}}(\boldsymbol{k})$. In addition, $\omega^{\text{marginal}}, \omega^{\text{joint}} \in \mathbb{R}^{j \times i}$ for $i, j = 1, \dots, N$ are 2-dimensional matrices that includes the pairwise weights between each position of the sequence and the rest of the positions. The weights are initialized uniformly across a certain sequence length dimension. For experiments related to language modeling, we can leverage information from attention mask to initialize the weights on the unmasked part of sequence. To speed up the computation for Transformer-RKDE, we use a single-step iteration on

Table 1: Perplexity (PPL) and negative likelihood loss (NLL) of our methods and baselines on WikiText-103 dataset. The best results are highlighted in bold font and the second best results are highlighted in underline. Transformer-RKDE and Transformer-SPKDE achieve competitive performance to the baseline methods while shows much better PPL and NLL under random swap with outlier words.

| Method | Clean Data | | Word Swap | |
|---|---|---|---|---|
| | Valid PPL/Loss | Test PPL/Loss | Valid PPL/Loss | Test PPL/Loss |
| Standard Softmax | 33.52/3.51 | 34.59/3.54 | 72.28/4.45 | 74.56/4.53 |
| Transformer-KDE | 33.34/3.51 | 34.37/3.54 | 71.94/4.43 | 73.75/4.49 |
| Transformer-RKDE (Huber) | 33.22/3.50 | 34.29/3.54 | 52.14/3.92 | 55.68/3.99 |
| Transformer-RKDE (Hampel) | 33.24/3.50 | 34.35/3.54 | 55.61/3.98 | 57.92/4.03 |
| Transformer-SPKDE | **33.05/3.49** | **34.18/3.53** | **51.36/3.89** | **54.97/3.96** |

equation 15 to approximate the optimal set of weights. Empirical results have shown that this one-step iteration can achieve sufficiently accurate results. For Transformer-SPKDE, we find the optimal set of weights via the QP solver. This strategy is shown to be effective during the empirical evaluation on both image and text data. The procedure of computing the attention vector for Transformer-RKDE and Transformer-SPKDE can be found at Algorithm 1.

## 4 EXPERIMENTAL RESULTS

In this section, we empirically validate the advantage of our proposed transformer integrated with robust KDE attention (Transformer-RKDE/SPKDE) over the standard softmax transformer and its nonparametric regression variant (Transformer-KDE in equation 9) on two large-scale datasets: language modeling on WikiText-103 dataset (Merity et al., 2016) (Section 4.1) and image classification on Imagenet (Russakovsky et al., 2015; Deng et al., 2009) and Imagenet-C (Hendrycks & Dietterich, 2019) (Section 4.2). Our experiments have shown that: (1) Transformer with robust KDE attention can reach competitive performance with baseline methods on a variety of tasks with different data modalities, this can be achieved without modifying the model architecture or introducing extra parameters; (2) the advantage of Transformer with robust KDE attention is more prominent when there is contamination of samples in either text or image data. All of our experiments are performed on the NVIDIA A-100 GPUs. For each experiment, we compare Transformer-RKDE/SPKDE with other baselines under the same hyper-parameter configurations. The implementation to reproduce our results can be found at anonymous.4open.science/r/robust-transformer-D7AB/README.md.

### 4.1 ROBUST LANGUAGE MODELING

**Dataset:** WikiText-103 is a language modeling dataset that contains collection of tokens extracted from good and featured articles from Wikipedia, which is suitable for models that can leverage long-term dependencies. The dataset contains around $268K$ words and its training set consists of about $28K$ articles with $103M$ tokens, this corresponds to text blocks of about 3600 words. The validation set and test sets consist of 60 articles with $218K$ and $246K$ tokens respectively. We follow the standard configurations in Merity et al. (2016); Schlag et al. (2021) and splits the training data into $L$-word independent long segments. During evaluation, we process the text sequence using a sliding window of size $L$ and feed into the model with a batch size of 1. The last position of the sliding window is used for computing perplexity except in the first segment, where all positions are evaluated as in Al-Rfou et al. (2019); Schlag et al. (2021).

**Implementation Details:** We used the language models developed by Schlag et al. (2021) in our experiments. The dimensions of key, value, and query are set to 128, and the training and evaluation context length are set to 256. As for self-attention, we set the number of heads as 8, the dimension of feed-forward layer as 2048, and the number of layers as 16. To avoid numerical instability, we apply the `log-sum-exp` trick in equation 9 when computing the attention probability vector through the Gaussian kernel. We apply similar tricks when computing the weights of KIRWLS algorithm, where we first obtain the weights in `log` space, followed by the `log-sum-exp` trick to compute robust self-attention as in equation 18.

Table 2: Top-1, top-5 accuracy (%) and mean corruption error (mCE) of DeiT with different attentions. The best results are highlighted in bold font and the second best are highlighted in underline. RVT (Mao et al., 2022) achieves better results on clean data and corrupted imagenet; DieT with robust KDE attention achieve better results under different adversarial attacks while still achieve competitive performance on corrupted imagenet.

| Method | Clean Data | | FGSM | | PGD | | SPSA | | Imagenet-C | |
|---|---|---|---|---|---|---|---|---|---|---|
| | Top 1 | Top 5 | Top 1 | Top 5 | Top 1 | Top 5 | Top 1 | Top 5 | Top 1 | mCE$\downarrow$ |
| Baseline DeiT | 72.23 | 91.13 | 52.61 | 82.26 | 41.84 | 76.49 | 48.34 | 79.36 | 42.38 | 71.14 |
| RVT | **74.37** | **93.89** | 53.67 | 84.11 | 43.39 | 77.26 | 51.43 | 80.98 | **45.64** | **68.57** |
| DeiT-KDE | 72.58 | 91.34 | 52.25 | 81.52 | 41.38 | 76.41 | 48.61 | 79.68 | 42.63 | 70.78 |
| DeiT-RKDE (Huber) | 72.83 | 91.44 | 55.83 | 85.89 | 44.15 | 79.06 | 52.42 | 82.03 | 45.58 | 68.69 |
| DeiT-RKDE (Hampel) | 72.94 | 91.63 | 55.92 | 85.97 | 44.23 | 79.16 | 52.48 | 82.07 | 45.61 | 68.67 |
| DeiT-SPKDE | 73.22 | 91.95 | **56.03** | **86.12** | **44.51** | **79.47** | **52.64** | **82.33** | 44.76 | 69.34 |

**Results:** In Table 1, we report the validation and test PPL of Transformer-RKDE (with Huber and Hampel loss functions), Transformer-RKDE versus the softmax transformer and its nonparametric regression variant. Based on the derivation in equation 11, we would expect Transformer-KDE to have similar performance with softmax transformer. Meanwhile, Transformer-RKDE and SPKDE is able to improve baselines PPL and NLL in both validation and test sets.

We can observe more obvious improvement when the dataset is under a word swap attack, which randomly replace selected keywords of input data by a generic token "$AAA$" during evaluation. Our method, particularly SPKDE-based robust attention, achieves much better results for down-weighting rare words, and therefore more robust to such kind of attack. Our implementation on word swap is based on the public code TextAttack by Morris et al. (2020)[1], while we use the greedy search method with the constraints on stop-words modification from the TextAttack library.

## 4.2 Image Classification under Adversarial Attack

**Dataset:** We use the full ImageNet dataset that contains $1.28M$ training images and $50K$ validation images. The model learns to predict the class of the input image among 1000 categories. We report the top-1 and top-5 accuracy on all experiments. For robustness on common image corruptions, we use ImageNet-C (Hendrycks & Dietterich, 2019) which consists of 15 types of algorithmically generated corruptions with five levels of severity. ImageNet-C uses the mean corruption error (mCE) as metric, while the smaller mCE means the more robust of the model under corruptions.

**Implementation Details:** Our method uses the same training configurations as DeiT-Tiny (Touvron et al., 2021b). Given that all approaches do not modify the model architecture, each employed model has $5.7M$ parameters. We also implemented a state-of-the-art robust vision transformer (RVT) model (Mao et al., 2022) as a baseline. For a fair comparison, we only implemented its position-aware attention scaling without further modifications on model architecture. The resulting model has around $7.2M$ parameters. To evaluate adversarial robustness, we apply adversarial examples generated by untargeted white-box attacks including single-step attack method FGSM (Goodfellow et al., 2014), multi-step attack method PGD (Madry et al., 2017) and score-based black-box attack method SPSA (Uesato et al., 2018). The attacks are applied on $100\%$ of the validation set of ImageNet. Both these attacks perturb the input image with perturbation budget $\epsilon = 1/255$ under $l_\infty$ norm; while PGD attack uses 20 steps with step size $\alpha = 0.15$.

**Results:** We summarize the results in Table 2. RVT achieves better performance on clean and corrupted imagenet. The set of DeiT with robust KDE attention can also obtain very close results with RVT under these settings while leading to much better results under different adversarial attacks. Figure 2 shows the relationship between accuracy versus perturbation budget using three attack methods. Our proposed methods can improve the accuracy under different perturbation budget and exhibits greater advantage with higher perturbation strength. We provide more ablation studies in Appendix B regarding to different design choices of the proposed robust KDE attention.

---

[1]Implementation available at github.com/QData/TextAttack

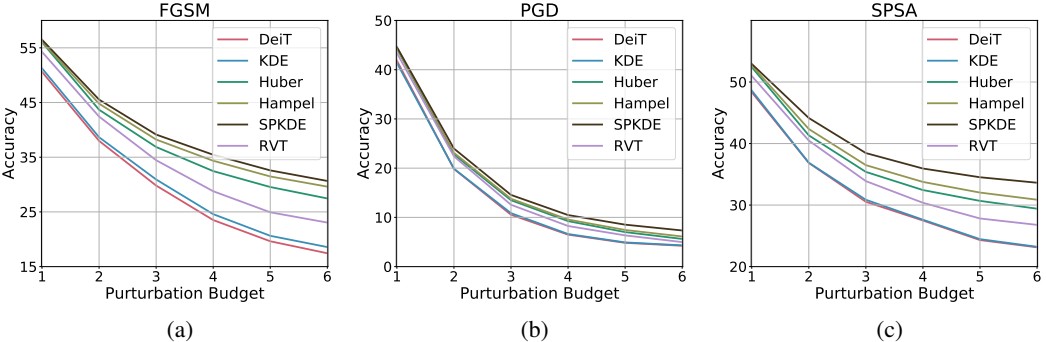

(a)            (b)            (c)

Figure 2: The top-1 classification *accuracy v.s. perturbation budget* $\times 255$ curves on ImageNet against three untargeted attack methods under the $l_\infty$ norm. Among all the competing methods, DeiT with robust KDE attention models show better robustness under all attack methods with different perturbation budgets.

## 5   RELATED WORKS

**Robustness of Transformer:** Vision Transformer (ViT) models (Dosovitskiy et al., 2020; Touvron et al., 2021b) recently achieved exemplary performance on a variety of vision tasks that can be used as a strong alternative to CNNs. To ensure its generalization ability on different datasets, many works (e.g., Subramanya et al., 2022; Paul & Chen, 2022; Bhojanapalli et al., 2021) have studied the robustness of ViT under different types of attacks. Mahmood et al. (2021) empirically shows that ViT is vulnerable to white-box adversarial attack but a simple ensemble defense can achieve unprecedented robustness without sacrificing clean accuracy. Mao et al. (2022) performs robustness analysis on different building blocks of ViT and proposed position-aware attention scaling and patch-wise augmentation that improved robustness and accuracy of ViT models. More recently, Zhou et al. (2022) proposed fully attentional networks to improve the self-attention and achieved state-of-the-art accuracy on corrupted images. However, these works focus on improving the architectural design of ViT targeted for some specific tasks, which lacks a general framework on improving the robustness of transformers. In addition, most of the recent works studying robustness of transformer concentrate on vision related tasks and cannot generalize across different data modalities.

**Theoretical Frameworks of Attention Mechanisms:** Attention mechanisms in transformers have been recently studied from different perspectives. Tsai et al. (2019) shows that attention can be derived from smoothing the inputs with appropriate kernels. Katharopoulos et al. (2020); Choromanski et al. (2021); Wang et al. (2020) further linearize the softmax kernel in attention to attain a family of efficient transformers with both linear computational and memory complexity. These linear attentions are proven in Cao (2021) to be equivalent to a Petrov-Galerkin projection (Reddy, 2004), thereby indicating that the softmax normalization in dot-product attention is sufficient but not necessary. Other frameworks for analyzing transformers that use ordinary/partial differential equations include Lu et al. (2019); Sander et al. (2022). In addition, the Gaussian mixture model and graph-structured learning have been utilized to study attentions and transformers (Tang & Matteson, 2021; Gabbur et al., 2021; Zhang & Feng, 2021; Wang et al., 2018; Shaw et al., 2018; Kreuzer et al., 2021).

## 6   CONCLUSION AND FUTURE WORKS

In this paper, via the connection between the dot-product self-attention mechanism in transformer with nonparametric kernel regression problem, we developed a family of robustified transformers by leveraging robust kernel density estimation as a replacement of dot-product attention to alleviate the effect from outliers. We show that the optimal estimation of potentially contaminated density functions via robust KDE requires computing a set of weights, which can be flexibly integrated when computing attentions in commonly used transformer models. Empirical evaluations have shown that Transformer-RKDE can improve performance on clean data while demonstrate robust results under various attacks on both vision and language modeling tasks. The robust KDE attention we developed has the merit of generalizing to the whole family of transformer models, which we intended to demonstrate as a future work. Meanwhile, we will also investigate better and more efficient approach to estimate the set of weights for robust kernel density estimations.

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

# Supplementary Material of "Robustify Transformers with Robust Kernel Density Estimation"

## A  PROOF OF PROPOSITION

**Proposition 2.** *Assume the robust loss function $\rho$ is non-decreasing in $[0, \infty]$, $\rho(0) = 0$ and $\lim_{x \to 0} \frac{\rho(x)}{x} = 0$. Define $\psi(x) := \frac{\rho'(x)}{x}$ and assume $\psi(0) = \lim_{x \to 0} \frac{\rho'(x)}{x}$ exists and finite. Then the optimal $\hat{p}_{robust}$ can be written as*

$$\hat{p}_{robust} = \sum_{j \in [N]} \omega_j k_\sigma(\boldsymbol{x}_j, \cdot),$$

*where $\omega = (\omega_1, \cdots, \omega_N) \in \Delta_N$, and $\omega_j \propto \psi\left(\|k_\sigma(\boldsymbol{x}_j, \cdot) - \hat{p}_{robust}\|_{\mathcal{H}_{k_\sigma}}\right)$. Here $\Delta_n$ denotes the $n$-dimensional simplex.*

*Proof.* The proof of Proposition 2 is mainly adapted from the proof in Kim & Scott (2012). Here, we provide proof of completeness. For any $p \in \mathcal{H}_{k_\sigma}$, we denote

$$J(p) = \frac{1}{N} \sum_{j \in [N]} \rho\left(\|k_\sigma(\boldsymbol{x}_j, \cdot) - p\|_{\mathcal{H}_{k_\sigma}}\right).$$

Then we have the following lemma regarding the Gateaux differential of $J$ and a necessary condition for $\hat{p}_{robust}$ to be optimal solution of the robust loss objective function in equation 13.

**Lemma 1.** *Given the assumptions on the robust loss function $\rho$ in Proposition 2, the Gateaux differential of $J$ at $p \in \mathcal{H}_{k_\sigma}$ with incremental $h \in \mathcal{H}_{k_\sigma}$, defined as $\delta J(p; h)$, is*

$$\delta J(p; h) := \lim_{\tau \to 0} \frac{J(p + \tau h) - J(p)}{\tau} = -\langle V(p), h \rangle_{\mathcal{H}_{k_\sigma}},$$

*where the function $V : \mathcal{H}_{k_\sigma} \to \mathcal{H}_{k_\sigma}$ is defined as:*

$$V(p) = \frac{1}{N} \sum_{j \in [N]} \psi\left(\|k_\sigma(\boldsymbol{x}_j, \cdot) - p\|_{\mathcal{H}_{k_\sigma}}\right) (k_\sigma(\boldsymbol{x}_j, \cdot) - p).$$

*A necessary condition for $\hat{p}_{robust}$ is $V(\hat{p}_{robust}) = 0$.*

The proof of Lemma 1 can be found in Lemma 1 of Kim & Scott (2012). Based on the necessary condition for $\hat{p}_{robust}$ in Lemma 1, i.e., $V(\hat{p}_{robust}) = 0$, we have

$$\frac{1}{N} \sum_{j \in [N]} \psi\left(\|k_\sigma(\boldsymbol{x}_j, \cdot) - \hat{p}_{robust}\|_{\mathcal{H}_{k_\sigma}}\right) (k_\sigma(\boldsymbol{x}_j, \cdot) - \hat{p}_{robust}) = 0.$$

Direct algebra indicates that $\hat{p}_{robust} = \sum_{j \in [N]} \omega_j k_\sigma(\boldsymbol{x}_j, \cdot)$ where $\omega = (\omega_1, \cdots, \omega_N) \in \Delta_N$, and $\omega_j \propto \psi\left(\|k_\sigma(\boldsymbol{x}_j, \cdot) - \hat{p}_{robust}\|_{\mathcal{H}_{k_\sigma}}\right)$. As a consequence, we obtain the conclusion of the proposition. $\square$

## B  ABLATION STUDIES

Table 3: Text PPL/NLL loss versus the parameter $a$ of Huber loss function defined in equation 14 (upper) and Hampel loss function (Kim & Scott, 2012) (lower; we use $2 \times a$ and $3 \times a$ as parameters $b$ and $c$) on original and word-swapped Wiki-103 dataset. The best results are highlighted in bold font and the second best are highlighted in underline. We choose $a = 0.4$ in rest of the experiments.

| Robust Loss Parameter | 0.1 | 0.2 | 0.4 | 0.6 | 0.8 | 1 |
|---|---|---|---|---|---|---|
| Clean Data | 34.92/3.57 | 34.87/3.56 | **34.29/3.54** | 34.38/3.54 | 34.46/3.54 | 34.48/3.54 |
| Word Swap | 56.82/4.01 | 55.97/3.99 | **55.68/3.99** | 57.89/4.03 | 58.26/4.04 | 58.37/4.04 |
| Clean Data | 34.67/3.55 | **34.32/3.54** | 34.35/3.54 | 34.47/3.54 | 34.53/3.54 | 34.58/3.54 |
| Word Swap | 58.02/4.03 | **57.86/4.03** | 57.92/4.03 | 58.24/4.04 | 58.37/4.04 | 58.43/4.04 |

Table 4: Top-1 classification accuracy on ImageNet versus the parameter $a$ of Huber loss function defined in equation 14 under different settings. The best results are highlighted in bold font and the second best are highlighted in underline. We choose $a = 0.2$ in rest of the experiments.

| Huber Loss Parameter | 0.1 | 0.2 | 0.4 | 0.6 | 0.8 | 1 |
|---|---|---|---|---|---|---|
| Clean Data | 71.45 | **72.83** | 71.62 | 71.07 | 70.65 | 70.34 |
| FGSM | **56.72** | 55.83 | 55.34 | 54.87 | 54.02 | 52.98 |
| PGD | **46.37** | 44.15 | 43.87 | 43.25 | 42.69 | 41.96 |
| SPSA | 52.38 | **52.42** | 51.69 | 51.34 | 50.97 | 48.22 |
| Imagenet-C | 45.37 | 45.58 | **45.63** | 45.26 | 44.63 | 43.76 |

Table 5: Top-1 classification accuracy on ImageNet versus the parameter $a$ of Hampel loss function defined in Kim & Scott (2012) under different settings. We use $2 \times a$ and $3 \times a$ as parameters $b$ and $c$. The best results are highlighted in bold font and the second best are highlighted in underline. We choose $a = 0.2$ in rest of the experiments.

| Hampel Loss Parameter | 0.1 | 0.2 | 0.4 | 0.6 | 0.8 | 1 |
|---|---|---|---|---|---|---|
| Clean Data | 71.63 | **72.94** | 71.84 | 71.23 | 70.87 | 70.41 |
| FGSM | **56.42** | 55.92 | 55.83 | 55.66 | 54.97 | 53.68 |
| PGD | **45.18** | 44.23 | 43.89 | 43.62 | 43.01 | 42.34 |
| SPSA | **52.96** | 52.48 | 52.13 | 51.46 | 50.92 | 50.23 |
| Imagenet-C | 44.76 | 45.61 | 46.04 | **46.13** | 45.82 | 45.31 |

Table 6: Top-1 classification accuracy on ImageNet versus the parameter $\beta$ of SPKDE defined in equation 16 under different settings. $\beta = \frac{1}{1-\varepsilon} > 1$, where $\varepsilon$ is the percentage of anomalous samples. A larger $\beta$ indicates a more robust model. The best results are highlighted in bold font and the second best are highlighted in underline. We choose $\beta = 1.4$ in rest of the experiments.

| $\beta$ | 1.05 | 1.2 | 1.4 | 1.6 | 1.8 | 2 |
|---|---|---|---|---|---|---|
| Clean Data | **74.25** | 73.56 | 73.22 | 73.01 | 72.86 | 72.64 |
| FGSM | 53.69 | 55.08 | **56.03** | 55.37 | 54.21 | 53.86 |
| PGD | 42.31 | 43.68 | **44.51** | 44.32 | 44.17 | 43.71 |
| SPSA | 51.29 | 52.02 | 52.64 | **52.84** | 52.16 | 51.39 |
| Imagenet-C | 44.68 | **45.49** | 44.76 | 44.21 | 43.96 | 43.33 |

Table 7: Top-1 classification accuracy on ImageNet versus the number of iterations of the KIRWLS algorithm in equation 15 employed in Transformer-RKDE. Since the increased number of iterations does not lead to significant improvements of performance while the computational cost is much higher, we use the single-step iteration of the KIRWLS algorithm in Transformer-RKDE.

| | Huber Loss | | | | Hampel Loss | | | |
|---|---|---|---|---|---|---|---|---|
| Iteration # | 1 | 2 | 3 | 5 | 1 | 2 | 3 | 5 |
| Clean Data | 72.83 | 72.91 | 72.95 | 72.98 | 72.94 | 72.99 | 73.01 | 73.02 |
| FGSM | 55.83 | 55.89 | 55.92 | 55.94 | 55.92 | 55.96 | 55.97 | 55.99 |
| PGD | 44.15 | 44.17 | 44.17 | 44.18 | 44.23 | 44.26 | 44.28 | 44.31 |
| SPSA | 52.42 | 52.44 | 52.45 | 52.45 | 52.48 | 52.53 | 52.55 | 52.56 |
| Imagenet-C | 45.58 | 45.61 | 45.62 | 45.62 | 45.61 | 45.66 | 45.68 | 45.71 |

Table 8: Computation time (measured by seconds per iteration) of baseline methods, Transformer-SPKDE and Transformer-RKDE with different number of KIRWLS iterations. Transformer-SPKDE requires longer time since it directly obtains the optimal set of weights via the QP solver.

| | Iterations of KIRWLS | | | | DeiT | RVT | SPKDE |
|---|---|---|---|---|---|---|---|
| | 1 | 2 | 3 | 5 | | | |
| Time (s/it) | 0.43 | 0.51 | 0.68 | 0.84 | 0.35 | 0.41 | 1.45 |

