# OpenReview forum: "Robustify Transformers with Robust Kernel Density Estimation"
_ICLR.cc/2023/Conference — Submitted to ICLR 2023_

### Official Review · Reviewer_cLTJ · 2022-10-19

**Confidence:** 4
**Clarity, Quality, Novelty And Reproducibility:** I fail to identify the original of th…
**Correctness:** 3
**Technical Novelty And Significance:** 1
**Empirical Novelty And Significance:** 1
**Recommendation:** 3

**Strength And Weaknesses:**

Strength:
1. The problem and issues to address are well motivated.

Weakness:
1. The paper appears to be largely overlapped with the existing literature of Kim and Scott (2012), which is intensively cited in the manuscript.
2. The robust self-attention mechanism does not have strong theoretical support

**Summary Of The Paper:**

The paper advocates to use robust kernel density estimation in the self-attention mechanism to mitigate the issue of data contamination. In particular, the idea is to down-weight the weight of bad samples in the estimation process. The authors conduct empirical experiments to demonstrate the effectiveness of the proposal.

**Summary Of The Review:**

I fail to identify the technical contribution of this manuscript over existing literature.

---

> ### Author Response · Authors · 2022-11-19
> **Response to Reviewer cLTJ**
>
> Dear Reviewer `cLTJ`,
>
> Thank you so much for your review! We believe that our work takes a different view of the robustness problem of Transformers in the RKHS domain, and is able to build the connections between the family of transformer models and robust nonparametric regression methods. We have extended our current approach under the nonparametric regression framework and implemented the scaling and projection KDE (SPKDE) for a robust attention mechanism. Along this direction, we will work on incorporating more state-of-the-art models. Please check our first post and revised paper for more discussion and experimental results.
>
> Sincerely,
>
> Authors

---

### Official Review · Reviewer_NtCQ · 2022-10-23

**Confidence:** 4
**Correctness:** 4
**Technical Novelty And Significance:** 3
**Empirical Novelty And Significance:** 2
**Recommendation:** 5

**Clarity, Quality, Novelty And Reproducibility:**

The presentation is clear and well-organized. The contributions are significant and somewhat new. The authors provide the code to reproduce their results.

**Strength And Weaknesses:**

Strengths:
+ The idea seems interesting. Starting from the inherent problem of self-attention (the KDE is not robust to the outliers), the authors introduce RKDE, the robust version of the KDE, to construct robust attention in transformer.
+ The experiments verify the effectiveness of the proposed model in both language modeling and image classification tasks.

Weaknesses:
+ Lack of quantitative analysis on the efficiency of the proposed model. Since additional weight calculation requires iteration, I wonder whether it will affect the efficiency of the algorithm.
+ Because the experimental verification is mainly designed for different adversarial attacks, I wonder how the final effect of the proposed method is compared with other models designed for adversarial attacks. This can better show the significance of the proposed method.


**Summary Of The Paper:**

This paper presents a robust variant of the standard Transformer named Transformer-RKDE to improve the robustness of data with contaminated samples. The idea is based on the interpretation that the self-attention in transformer can be viewed as a non-parametric estimator based on the kernel density estimation (KDE). The authors re-interpret KDE as a regression in the Reproducing Kernel Hilbert Space (RKHS) and indicate that the vanilla KDE is sensitive to data corruption. Therefore, the authors introduce a robust version of the KDE and use the robust KDEs to construct the robust self-attention mechanism for the Transformer.

**Summary Of The Review:**

Overall, the innovation is somewhat interesting to me. However, I have some concerns about its efficiency and practical application value.

---

> ### Author Response · Authors · 2022-11-19
> **Response to Reviewer NtCQ**
>
> Dear Reviewer `NtCQ`,
>
> Thank you for your insightful comments and kind review! Regarding to your concerns on efficiency of KIRWLS algorithm and baseline methods, we have added additional experiments on (1) The number of iterations of KIRWLS algorithm used in Transformer-RKDE and its impact on model performance and computational cost. (2) State-of-the-art robust vision transformer (RVT) model as a new baseline for our experiments on image data. Please check our revised paper and the first post for more experimental results.
>
> Sincerely,
>
> Authors

---

> > ### Comment · Reviewer_NtCQ · 2022-12-07
> > **Thanks for the response.**
> >
> > Thanks for the response. After checking the comments of other reviewers and the author's reply, the reviewers' concerns have not been well addressed, so I decided to revise the score to 5.

---

### Official Review · Reviewer_N1gC · 2022-10-23

**Confidence:** 4
**Correctness:** 3
**Technical Novelty And Significance:** 2
**Empirical Novelty And Significance:** 2
**Recommendation:** 5

**Clarity, Quality, Novelty And Reproducibility:**

The quality of this paper is good, the problem is clearly stated, but the novelty seems limited.

**Strength And Weaknesses:**

strengths:
1. propose a novel robust transformer framework which replace the dotproduct attention by an attention arising from the robust kernel density estimators (RKDE) associated with the robust kernel regression problem.
2. the proposed method only requires computing an extra set of weights.

weaknesses:
1. My most critical concern is regarding the review of the related works. Several robust transformer methods have been proposed, I suggest that the author should review those works in the related works, please see an incomplete list here, Mao X, Qi G, Chen Y, et al. Towards robust vision transformer[C]//Proceedings of the IEEE/CVF Conference on Computer Vision and Pattern Recognition. 2022: 12042-12051. Li N, Liu Y, Wu Y, et al. Robutrans: A robust transformer-based text-to-speech model[C]//Proceedings of the AAAI Conference on Artificial Intelligence. 2020, 34(05): 8228-8235. Liu J, Singhal T, Blessing L T M, et al. Crisisbert: a robust transformer for crisis classification and contextual crisis embedding[C]//Proceedings of the 32nd ACM Conference on Hypertext and Social Media. 2021: 133-141. Yang J, Gupta A, Upadhyay S, et al. TableFormer: Robust Transformer Modeling for Table-Text Encoding[J]. arXiv preprint arXiv:2203.00274, 2022.
2. The second concern is the baselines. The current baseline methods seem insufficient. Why does the author choose these methods as the baseline? Why does comparison with these methods can prove the effectiveness of the proposed method? I think the authors should select some similar methods (see weaknesses 1) as a baseline to prove the effectiveness of the proposed method.



**Summary Of The Paper:**

The authoor leverage the robust kernel density estimation (RKDE) in the self-attention mechanism, to alleviate the issue of the contamination of data by down-weighting the weight of bad samples in the estimation process.  Empirical results on language modeling and image classification tasks have demonstrated the effectiveness of this approach. The motivation is clear, but still have few concerns.

**Summary Of The Review:**

Motivation of this paper is clear, but the reviwe of the related works and baselines are insufficient.

---

> ### Author Response · Authors · 2022-11-19
> **Response to Reviewer N1gC**
>
> Dear Reviewer `N1gC`,
>
> We thank you for your helpful suggestions! We have incorporated the related works you mentioned into the paper and added an additional baseline that is most relevant to us according to your comments. Note that, to the best of our knowledge, we are the first to study the robustness problem in language modeling, and we believe that our method can serve as a reference for future research. Since the time of the rebuttal period is short, we are willing to explore and add empirical evaluations on other tasks such as text-to-speech and crisis classification.
>
> Sincerely,
>
> Authors

---

> > ### Comment · Reviewer_N1gC · 2022-12-07
> > **novelty and performance**
> >
> > Thanks for the response. However, the novelty is still limited, and I'm not sure about the performance of the proposed method regarding the added baselines.
> > Furthermore, I do not see any different contribution between the proposed method and those related works, which is very important.

---

### Official Review · Reviewer_tAEV · 2022-10-26

**Confidence:** 2
**Correctness:** 3
**Technical Novelty And Significance:** 2
**Empirical Novelty And Significance:** 2
**Recommendation:** 3

**Clarity, Quality, Novelty And Reproducibility:**

Clarity: The current presentation could be improved.

Quality: Based on the current presentation, the overall quality doesn't meet the bar of the top ML conference like ICLR.

Novelty: This paper considered an interesting attention variant motivated by robust kernel estimation.

Reproducibility: The code is provided.

**Strength And Weaknesses:**

Strength:
The proposed method enjoys promising numerical performance when the noise level is high.

Weakness:

1.  The writing could be significantly improved. For example, equations (5-11) in subsection 2.2 on page 3 are almost analogous to equations (3-8) in Nguyen et al. (2022). It may not necessary to use a whole page to restate the existing result in the literature.
Moreover, in Aglrotihm 1 on page 6, a) the $\hat{p}_{\mathrm{robust}}^{(k)}(k)$ is used without declaration, in the input
line the initial weights of $\omega\_{\mathrm{marginal}}^{(0)}$ and $\omega\_{\mathrm{joint}}^{(0)}$  are not used, and I don't find the definition of $\omega_j^{\mathrm{joint}}$.

2. The bond between theoretical motivation and real implementation is kind of weak. For example, the theoretical analysis requires the solve for $\hat{p}_{\mathrm{robust}}^{(k)}$ via an iterative method (e.g., KIRWLS). However, when checking the code, the RBFMultiHeadAttn class in Line 391 of src/mem_transformer.py, I don't find the for loop in the forward function and it seems the attention
being computed via only a single path...  If it is the case, the $\hat{p}\_{\mathrm{robust}}^{(k)}$ can be far away from its underlying *optimal* value. If it is not the case, I suggest authors report the computing cost of the proposed method since the multi-round of the iterative method can increase the computation time by a significant amount.



**Summary Of The Paper:**

This paper proposed a robustified transformer model motivated by the robust kernel density estimation. The proposed model alleviates the influence of bad data (e.g., outliers). The empirical studies on NLP and CV tasks demonstrate the effectiveness of the proposed methods.

**Summary Of The Review:**

This paper proposes a variant of Transformer motivated by robust kernel estimation. In particular, a decayed weight $\omega$ is computed to minimize the influence of the outliers, and in high noise settings, the proposed method reaches promising numerical performance.

My major concern is the writing of the paper which could be significantly improved. I also suggest authors to further highlight the computation cost of the iterative approach or show the single path can yield an accurate enough estimation of $\hat{p}_{\mathrm{robust}}^{(k)}$.


Minor Issue: I suggest authors use \eqref instead of \ref when referring to the equation number.


At the current stage, I tend to reject this paper but I'm willing to change my evaluation after rebuttal.

---

> ### Author Response · Authors · 2022-11-19
> **Response to Reviewer tAEV**
>
> Dear Reviewer `tAEV`,
>
> We are appreciative of your writing suggestions and helpful feedback! We have corrected our notations/definitions based on your comments. We have made attempt to shorten Subsection 2.2, while we are also trying to avoid significant changes to the overall paper layout since this subsection is essential in our following demonstration. In addition, we have added additional analysis on the number of iterations of the KIRWLS algorithm used in Transformer-RKDE and its impact on model performance and computational cost. Please check our first post and revised paper for the results and discussions.
>
> Sincerely,
>
> Authors

---

### Official Review · Reviewer_3QyS · 2022-10-31

**Confidence:** 3
**Correctness:** 4
**Technical Novelty And Significance:** 3
**Empirical Novelty And Significance:** 2
**Recommendation:** 5

**Clarity, Quality, Novelty And Reproducibility:**

clarity, quality and reproducibility are good
novelty is weak


**Details Of Ethics Concerns:**

no concerns

**Strength And Weaknesses:**

Strength
1. The topic is interesting and important
2. The paper is clear to understand and easy to follow
3. Related works and backgrounds are well discussed and explained.
4. Strong experiments are shown the benefits of the proposed method

Weakness
1. The novelty is weak and lacks of a major contribution
2. The comparison is not strong and lacks of other robust transformer models instead of KDE-based models

**Summary Of The Paper:**

This paper develops transformer-RKDE by leveraging robust kernel density estimation as a replacement of dot-product attention. The authors show that the optimal estimation of density functions via robust KDE requires computing a set of weights by solving an iterative re-weighted least-square problem. Multiple experiments including NLP and vision demonstrate the proposed method on performance and efficiency improvement.

**Summary Of The Review:**

The paper simply employs the robust KDE (20212 JMRL) for transformer models. The idea is great and the connection is natural and straightforward but the major contribution seems marginal.

I also expect to see the performance benefits compared with other robust transformer models not only KDE-based vanilla models. That's important to demonstrate the necessity of the proposed robust KDE.

---

> ### Author Response · Authors · 2022-11-19
> **Response to Reviewer 3QyS**
>
> Dear Reviewer `3QyS`,
>
> Thank you so much for your helpful suggestions! We address your concerns below:
>
> [**Novelty of Paper**] We believe that our work takes a different view of the robustness problem of Transformers in the RKHS domain, and is able to build the connections between the family of transformer models and robust nonparametric regression methods. We have extended our current approach under the nonparametric regression framework and implemented the scaling and projection KDE (SPKDE) for a robust attention mechanism. Along this direction, we will work on incorporating more state-of-the-art models.
>
> [**Baselines**] Please check our first post and revised paper for the results and discussions of the updated baseline.
>
> Sincerely,
>
> Authors

---

### Author Response · Authors · 2022-11-19
**Paper Revision and Updated Experiments**

Dear Reviewers,

We are really appreciative of your insightful feedback and comments. We are delighted to see that Reviewers find the problem interesting and well-motivated (`3QyS`, `NtCQ`, `cLTJ`), the ideas are novel and easy to follow (`3QyS`, `N1gC`), and demonstrate clear advantages using strong experiments (`3QyS`, `tAEV`, `NtCQ`). Here, we first wish to notify reviewers of the revision of this paper on additional experiments and updates of methods to address some of your common concerns. The revised parts are highlighted in blue.


**Update of Method**

Our work takes a different view of the robustness problem of Transformers in the RKHS domain and is able to build connections between the family of transformer models and robust nonparametric regression methods. To strengthen this motivation, we further extend our method under the nonparametric regression framework and implemented the scaling and projection KDE (SPKDE) for a robust attention mechanism. We have updated SPKDE as part of the robust attention mechanism in **Section 3.2**. The procedure for constructing robust SPKDE attention is similar to the RKDE approach we proposed earlier.


**Additional Experiments**

[Baseline Method] We implemented the state-of-the-art robust vision transformer (RVT) by [1] as a new baseline for our experiments on image data. The updated results and discussions can be found in **Section 4.2**. To the best of our knowledge, we are the first to study the robustness problem in language modeling. We believe our method can serve as a reference for future research.

[Computational Cost] As recommended by reviewers tAEV and NtCQ, we provide further analysis on the number of iterations of the KIRWLS algorithm used in Transformer-RKDE and its impact on model performance and computational cost. The results can be found in **Appendix B**. From the results, we can find that multiple iterations of KIRWLS will not significantly improve the performance and therefore the one-step approximation is sufficient to provide accurate results.

[SPKDE] We add the results from the newly introduced method SPKDE. They can be found in both **Section 4.1** and **Section 4.2**. Essentially, SPKDE achieves better performance than RKDE in these experiments as it finds the optimal set of weights using QP solvers.

[Ablation Studies] We provide additional ablation studies on different design choices of our proposed method, including hyper-parameters and the type of robust loss function. They are highlighted in the experimental results.

Sincerely,

Authors

---

### Comment · Area_Chair_yxBy · 2022-11-20
**Please update your reviews**

Please make sure that your reviews acknowledge authors’ responses and reflect your current evaluation of the paper. This is particularly important if you didn’t directly engage with the authors during the discussion phase (so the authors don’t know if their response changed your evaluation) or if you expressed an intention to update your rating but did not do so.

Cheers,
AC

---

### Author Response · Authors · 2022-11-26
**Please let us know if you have any further questions**

Dear Reviewers,

We would like to thank you again for your insightful comments and feedback! We have attempted to address the concerns you raised and hopefully, they are helpful in evaluating our submission. Please let us know if you have any further questions and we are happy to answer them anytime. Thank you very much!

Sincerely,

Authors

---

### Decision · Program_Chairs · 2023-01-20

**Decision:**

Reject

**Justification For Why Not Higher Score:**

The experiment is insufficient and novelty is limited.

**Justification For Why Not Lower Score:**

NA

**Metareview: Summary, Strengths And Weaknesses:**

This paper presents a robust variant of the standard Transformer named Transformer-RKDE to improve the robustness of data with contaminated samples. The idea is based on the interpretation that the self-attention in transformer can be viewed as a non-parametric estimator based on the kernel density estimation (KDE).

Strengths: The problem and issues to address are well motivated. The idea is great and the connection is natural and straightforward.
Weakness: All the reviewers agree that the critical part is the limited novelty(simply employs the robust KDE for transformer model), concerns about its efficiency and practical application value. The reviewers are not satisfactory on the rebuttal since majority concerns are not well addressed.